# Pathogenesis and Therapy of Hermansky–Pudlak Syndrome (HPS)-Associated Pulmonary Fibrosis

**DOI:** 10.3390/ijms252011270

**Published:** 2024-10-19

**Authors:** Xiao Hu, Zhixiao Wei, Yumeng Wu, Manhan Zhao, Liming Zhou, Qiong Lin

**Affiliations:** School of Medicine, Jiangsu University, 301 Xuefu Road, Zhenjiang 212013, China; 2212213071@stmail.ujs.edu.cn (X.H.); 2212213070@stmail.ujs.edu.cn (Z.W.); 2212213095@stmail.ujs.edu.cn (Y.W.); 2212313096@stmail.ujs.edu.cn (M.Z.); 2212313098@stmail.ujs.edu.cn (L.Z.)

**Keywords:** Hermansky–Pudlak syndrome, pulmonary fibrosis, pathogenesis, lung AT2 cells, therapeutic strategies

## Abstract

Hermansky–Pudlak syndrome (HPS)-associated pulmonary fibrosis (HPS-PF) is a progressive lung disease that is a major cause of morbidity and mortality in HPS patients. Previous studies have demonstrated that the HPS proteins play an essential role in the biogenesis and function of lysosome-related organelles (LROs) in alveolar epithelial type II (AT2) cells and found that HPS-PF is associated with dysfunction of AT2 cells and abnormal immune reactions. Despite recent advances in research on HPS and the pathology of HPS-PF, the pathological mechanisms underlying HPS-PF remain poorly understood, and no effective treatment has been established. Therefore, it is necessary to refresh the progress in the pathogenesis of HPS-PF to increase our understanding of the pathogenic mechanism of HPS-PF and develop targeted therapeutic strategies. This review summarizes the recent progress in the pathogenesis of HPS-PF provides information about the current treatment strategies for HPS-PF, and hopefully increases our understanding of the pathogenesis of HPS-PF and offers thoughts for new therapeutic interventions.

## 1. Introduction

Hermansky–Pudlak syndrome (HPS) is a rare autosomal recessive disorder with an equal prevalence in both sexes [1]. It manifests as ocular or oculocutaneous albinism symptoms and hemorrhagic qualities, and, sometimes, is accompanied by other abnormalities, such as immune deficiencies, neurological symptoms, granulomatous colitis, and pulmonary fibrosis [2]. Since being first reported in 1959 [3], 11 distinct types of HPS genes, including *HPS1*, *AP3B1*, *HPS3*, *HPS4*, *HPS5*, *HPS6*, *DTNBP1*, *BLOC1S3*, *BLOC1S6*, *AP3D1*, and *BLOC1S5*, have been identified to date [4]. The products of these genes are the components of the biogenesis of the lysosome-related organelle complexes (BLOCs) and adaptor protein complex-3 (AP-3). Defects in these complexes appear to be the causes of HPS [5,6,7].

HPS has been discovered in many countries and continents worldwide, including China, Japan, Africa, the United States, and Spain [8,9,10]. The global prevalence of HPS has been estimated to be 1–9 per million [11], and according to the US Genetic and Rare Diseases Information Center, 30,000–200,000 people may have HPS [12], but the true frequency of HPS remains unknown. However, *HPS1* is the most prevalent subtype of HPS, accounting for approximately 50% of global HPS prevalence [13].

Pulmonary fibrosis is a lung disease caused by damage to the alveolar epithelial cells and the lung structure [14,15], dysregulation of lung immune reactions [16], and abnormal accumulation of extracellular matrix [17] and fibroblasts [17,18]. Thus, pulmonary fibrosis impairs the structure of alveoli, alters gas exchange, and ultimately leads to loss of lung function, respiratory failure, and even death [19]. Pulmonary fibrosis has been found in three genetic variants of HPS, including *HPS1*, *HPS2*, and *HPS4*. Studies have shown that HPS1 or HPS4 lung fibers are more prone to developing pulmonary fibrosis during middle age, between the ages of 30 and 50 [20,21], while HPS2 pulmonary fibrosis is more prevalent in children and young adults [22]. Notably, 100% of HPS1 patients have HPS-PF disease progression and ultimately die from HPS-PF [21].

HPS-associated pulmonary fibrosis (HPS-PF) is regarded as a major lethal factor in HPS, exhibiting several similarities to idiopathic pulmonary fibrosis (IPF) [19]. Distinctive characteristics of HPS-PF include the enlarged size of lamellar bodies (LBs) and foam-like degeneration of alveolar epithelial type II cells (AT2), as well as the presence of ceroid-like deposits in alveolar macrophages, which contain a considerable amount of surfactant and phospholipids [23], Currently, lung transplantation is the only option for the treatment of HPS-PF patients [24].

The pathogenesis of HPS-PF involves multiple factors, such as genetic background [25], lysosome-associated organelle defects [26], damage and repair of alveolar epithelial cells [14], immune dysregulation, inflammatory response [27], abnormal fibroblast growth [28], mitochondrial dysfunction [29], lysosomal and endoplasmic reticulum (ER) stress [30], autophagy abnormality [31], metabolic dysregulation, and aging [32]. Among them, apoptosis of AT2 cells and immune and inflammatory reactions play a major role in regulating the progression of HPS-PF.

Although some progress has been made in the study of HPS-PF phenotypes, the pathological mechanisms are still relatively superficial. Currently, there are no feasible preventive and therapeutic measures for this disease, and many questions remain unanswered. In order to accurately carry out targeted therapy, we need to clarify the connections between HPS-PF and its pathogenic mechanisms. Therefore, this review article focuses mainly on the pathogenesis of HPS-PF. Furthermore, we also discuss possible treatment strategies for HPS-PF and the challenges we may face in its treatment.

## 2. HPS and Lysosome-Related Organelles (LROs)

The HPS disease is caused by defects in the biogenesis, maturation, and secretion of LROs, which are rooted in the functional mutation of HPS genes [33]. LROs represent a functionally diverse group of cell type-specific organelles [34]. The main LROs identified to date include LBs of AT2 cells, melanosomes of pigmented cells, alpha and dense granules in platelets, Weibel–Palade bodies in endothelial cells, lytic granules in natural killer (NK) cells and cytotoxic T lymphocytes (CTLs), and basophilic and azurophilic granules in leukocytes [33,34,35]. The clinical manifestations of HPS, including hypopigmentation, platelet abnormalities, immune dysregulation, and pulmonary fibrosis, are aligned with a defect in the LROs [36].

BLOCs, including BLOC-1, BLOC-2, and BLOC-3, and AP-3 play a critical role in the biogenesis and maturation of LROs [37,38]. All the HPS subtypes identified are associated with dysfunction of BLOCs and AP-3. BLOCs and AP-3 are the protein complexes composed of HPS gene products. BLOC-1 has eight subunits, including BLOC1S1, BLOC1S2, HPS8 (BLOC1S3), BLOC1S4 (CNO or Cappuccino), HPS11 (BLOC1S5, MU or Muted), HPS9 (BLOC1S6, PLDN or Pallidin), BLOC1S7 (SNAPAP or Snapin), and HPS7 (BLOC1S8, DTNBP1 or Dysbindin) [39]; BLOC-2 contains three subunits, including HPS3, HPS5, and HPS6 [40]; BLOC-3 is composed of two subunits HPS1 and HPS4 [41]; AP-3 is a heterotetramer protein complex containing HPS2 (AP3B or β3), HPS10 (AP3D1 or δ), μ3, and σ3 subunits [4]. The defects in LROs and clinical phenotypes caused by dysfunction of the corresponding subunit in the complexes in each of the HPS subtypes are summarized in Table 1. Apparently, dysfunction of the subunit HPS7, HPS8, HPS9, or HPS11 in BLOC-1 is responsible for HPS subtype 7, 8, 9, or 11, respectively; the same is true for the subunits in BLOC-2, BLOC-3 and AP-3 (Table 1).

As shown in Table 1, HPS progressive pulmonary fibrosis is primarily associated with dysfunction of the subunits HPS1 and HPS4 in BLOC-3 and the subunit HPS2 in the AP-3 complex [38]. Thus, progressive pulmonary fibrosis in HPS is mainly caused by functional defects in the BLOC-3 or AP-3 complexes [42].
ijms-25-11270-t001_Table 1Table 1Protein complexes, LROs defects, and clinical phenotypes of HPS subtypes.Protein ComplexHPS GeneLROs DefectClinical Phenotype
BLOC-1*HPS7**(DTNBP1)*Melanosome,dense granule, Weibel–Palade bodyOcular skin albinism, bleeding diathesis, schizophrenia, nystagmus, reduced lung compliance[43,44,45]
*HPS8**(BLOC1S3)*Melanosome, dense granule, Weibel–Palade bodyModerate ocular skin albinism, mild bleeding diathesis, nystagmus[46,47]
*HPS9**(BLOC1S6)*Melanosome, dense granule, Weibel–Palade body, azurophilic granule, synaptic vesicleOcular skin albinism,mild bleeding diathesis,nystagmus, immunodeficiency, recurrent skin infections, [48,49]
*HPS11**(BLOC1S5)*Melanosome, dense granule, Weibel–Palade bodyMild ocular skin albinism, mild bleeding diathesis[4,50]BLOC-2*HPS3*Melanosome, dense granule, Weibel–Palade body, large dense core vesicleMild ocular skin albinism, mild bleeding diathesis, nystagmus, mild inflammatory bowel disease[51,52]
*HPS5*Melanosome, dense granule, Weibel–Palade bodyMild ocular skin albinism, nystagmus, mild bleeding diathesis, mild inflammatory bowel disease,hypercholesterolemia[53,54,55]
*HPS6*Melanosome, dense granule, Weibel–Palade body, large dense core vesicle, lamellar bodyOcular skin albinism, severe bleeding, nystagmus, respiratory and urinary tract infections, developmental delay and hearing loss, mild inflammatory bowel disease[52,56,57]BLOC-3*HPS1*Melanosome, dense granule, Weibel–Palade body, large dense core vesicle, lamellar bodyPulmonary fibrosis,ocular skin albinism, bleeding diathesis, interstitial lung disease, granulomatous colitis[8,26,58]
*HPS4*Melanosome, dense granule, Weibel–Palade body, large dense core vesicle, lamellar bodyPulmonary fibrosis,ocular skin albinism,bleeding diathesis,interstitial lung disease,granulomatous colitis,schizophrenia[59,60]AP-3*HPS2**(AP3B1)*Melanosome,dense granule, Weibel–Palade body, large dense core vesicle,MHC class II compartment, lamellar body,lytic granule,azurophilic granulePulmonary fibrosis,ocular skin albinism,bleeding diathesis,immunodeficiency,interstitial lung disease,mild facial dysmorphism,hip dysplasia,persistent hepatosplenomegaly,periodontitis,viral infection and Hodgkin’s lymphoma susceptibility,hemophagocytic lymphohistiocytosis,renal plate dysplasia,poor balance and hearing impairment[7,61]
*HPS10**(AP3D1)*Melanosome, dense granule, Weibel–Palade body, large dense core vesicle, MHC class II compartment, azurophilic granule, synaptic vesicleOcular skin albinism,immunodeficiency, susceptibility to airway infection,neutropenia,mild facial dysmorphism, hip dysplasia,persistent hepatosplenomegaly, neurodevelopmental delay,early onset seizures,hearing impairment[6,62]


## 3. Pathogenesis of HPS-PF

To date, the pathogenic mechanism of HPS-PF remains incompletely understood. Current studies indicate that the main pathogenic causes of HPS-PF are derived from dysfunctional alveolar epithelial cells, abnormal immune and inflammatory reactions, aberrant lung fibroblast proliferation, aberrant autophagy, metabolic derangement, and aging. The HPS-associated pathological factors are summarized in Figure 1.

### 3.1. Alveolar Epithelial Cells and HPS-PF

The alveolar epithelium consists of type I (AT1) and type II (AT2) cells. Alveolar epithelial type I (AT1) cells are large, thin, flattened squamous cells that cover most of the alveolar surface, take part in gas exchange, and maintain ionic and fluid homeostasis [63]. AT2 cells act as progenitor cells for self-renewal and transdifferentiation into AT1 cells [64], which are involved in lung homeostasis, repair, and regeneration [64,65,66]. In addition to the progenitor function, AT2 cells have multiple other cellular functions, such as secreting surfactant [67], maintaining fluid homeostasis [68], and regulating immune responses [69]. The homeostatic renewal and reparative regeneration of AT2 cells are essential for the maintenance of lung function, particularly during the development of pulmonary fibrosis, where AT2 cells play a dominant role [70]. Dysfunction of alveolar epithelial cells is directly associated with the pathogenesis of HPS-PF. The factors of alveolar epithelial cells associated with HPS-PF include (a) defects in LBs, the LROs in AT2 cells; (b) lysosomal and ER stress; (c) mitochondrial dysfunction and oxidative stress; (d) interaction of dysfunctional AT2 cells with macrophages; and (e) abnormal AP-3 signaling pathways in AT2 cells. These factors may accelerate the progression of HPS-PF.

#### 3.1.1. Defect in Lamellar Bodies (LBs) of AT2 Cells

Biogenesis and maturation of LBs in AT2 cells are dependent on HPS proteins [71,72]. LBs function for surfactant storage and secretion [73]. AT2 cells secrete LB-loaded surfactants to the alveolar lumen, thereby stabilizing the alveolar structure. Lung surfactants consist of a variety of lipids (primarily phosphatidylcholine; PC) and four distinct surfactant proteins (SP-A, -B, -C, and -D) [71]. Of these, SP-B and SP-C are hydrophobic proteins that play a significant role in reducing surface tension activity [74,75]. SP-A and SP-D are hydrophilic proteins that play a significant role in host defense mechanisms [76,77]. The formation of giant LBs in HPS1 patients and Hps1/Hps2 mutant mice has been found due to the abnormal accumulation of surfactants [78]. This accumulation is primarily composed of PC, SP-B, and SP-C [78,79]. This aberrant accumulation of surfactants has been found to be primarily associated with impaired secretion, with no significant relationship to abnormal transport or recycling [78,79]. Similarly, it has been observed that the deletion of peroxiredoxin 6 (PRDX6) in LBs of AP-3 deficient mice results in the accumulation of surfactant phospholipids in AT2 cells [80]. PRDX6 plays a pivotal role in maintaining laminar phospholipid homeostasis [81] and facilitating cell membrane repair following oxidative stress [82]. Transport of PRDX6 to LBs, an important step for the maturation of LBs, is dependent on AP-3 [80]. Thus, it is postulated that the mutation of HPS genes may lead to the blockage of LB maturation and the inability to secrete surfactant normally, which accelerates the apoptosis of AT2 cells, weakening the pulmonary repair capacity, and ultimately causing HPS-PF.

#### 3.1.2. Lysosomal and Endoplasmic Reticulum (ER) Stress Underlies Apoptosis in HPS1/2 Alveolar Epithelial Cells

ER stress has been associated with numerous diseases, including cancer, neurodegenerative diseases, and cardiovascular disease, Recently, it has also been linked to respiratory disease [83,84,85]. In IPF, ER stress has been found to be associated with apoptosis and epithelial-mesenchymal transition (EMT) in AT2 cells [86,87]. Moreover, a growing body of research indicates that lysosomal and ER stress is the basis of apoptosis in AT2 cells from Hps1/2 mice [30]. In AT2 cells of Hps1/2 mice with HPS-PF, extensive surfactant accumulation was observed [30]. The lysosomal stress occurred in the early stages of the disease, as manifested by elevated levels of cathepsin D [30]. In the late stages, these cells experienced ER stress, characterized by increased levels of the ER stress markers C/EBP homologous protein (CHOP; GADD153) and activating transcription factor-4 (ATF4) [88,89]. Previous studies have indicated that mutations in pro-surfactant protein C may trigger ER stress, dysfunction of the proteasome, and activation of cysteinyl asparaginase 3 [90]. In HPS-PF, the emergence of the ER stress response may be caused by increased accumulation of SP-B/C or reverse accumulation of lysosomal surfactant compounds into the ER, and this additional ER stress further exacerbates lysosomal stress-induced apoptosis in AT2 cells [30]. In addition, research in Hps mice and human samples has shown that cathepsin D not only facilitates apoptosis of alveolar epithelial cells but also stimulates the proliferation of fibroblasts [30,91,92]. These findings suggest that cathepsin D may play an important role in pulmonary fibrosis, and function as a key regulator in the development of HPS-PF.

In conclusion, surfactant abnormalities in HPS-PF cause lysosomal and endoplasmic reticulum stress, which leads to AT2 apoptosis and initiates the development of HPS-PF.

#### 3.1.3. Interaction of Dysfunctional AT2 Cells with Macrophages

It has been found that the single mutant Hps1 or Hps2 mice demonstrated increased fibrosis sensitivity but generally did not have spontaneous pulmonary fibrosis, whereas double mutant Hps1/2 mice exhibited spontaneous fibrosis with age [93,94]. Lung fibrosis in spontaneous Hps mice is caused by defects in intracellular transport, which reduce the apoptosis threshold of lung epithelial cells [93]. HPS fibrosis susceptibility and macrophage activation are due to AT2 cell dysfunction and have little to do with intrinsic defects in macrophage function [95]. In HPS-PF, dysfunctional AT2 cells are capable of producing excess monocyte chemoattractant protein-1 (MCP-1), which facilitates the recruitment and activation of transforming growth factor-β (TGF-β) in lung macrophages [96]. This, in turn, amplifies the fibrotic cascade response through apoptosis of AT2 cells and stimulation of fibrotic remodeling [96]. Furthermore, blocking the MCP-1/C-C chemokine receptor type 2 (CCR2) signaling pathway and the epithelial-specific transgenic correction on the HPS-deficiency significantly attenuated bleomycin-induced AT2 cells apoptosis, fibrosis susceptibility, and macrophage activation [95,96]. This suggests that the AT2 cells play a critical role in maintaining alveolar homeostasis and regulating alveolar macrophage activation. Furthermore, alveolar epithelial–macrophage interactions determine susceptibility to pulmonary fibrosis in HPS.

#### 3.1.4. Mitochondrial Dysfunction and Oxidative Stress

Some studies have highlighted the pivotal role of mitochondrial homeostasis disturbances in AT2 cells, fibroblasts, and alveolar macrophages in the pathogenesis of IPF [97]. Mitochondrial dysfunction has been identified in both a somatic cell model differentiated from HPS1 patient-specific human induced pluripotent stem cells (iPSCs) and HPS1 patient-specific alveolar epithelial cells [98]. It has been observed that PTEN-induced putative kinase 1 (PINK1) deficiency leads to mitochondrial swelling, dysfunction, and defective mitochondrial autophagy, which in turn promotes the HPS-PF process [97]. Mitochondrial dysfunction is a critical factor in the apoptosis of AT2 cells [99]. Therefore, recovery of mitochondrial dysfunction may be beneficial for the treatment of HPS-PF. Furthermore, increased reactive oxygen species (ROS) production and an unbalanced redox state have been associated with the activation of apoptosis [100]. ROS are thought to drive fibrotic remodeling by inducing epithelial damage and promoting lung inflammation and myofibroblast activation.

One study found that increased mitochondrial ROS levels were detected in the lung tissue of Hps1 mice [29]. It has been postulated that redox imbalance represents an early manifestation of HPS-PF and that antioxidants may be employed as an early therapeutic intervention for HPS-PF [29].

#### 3.1.5. AP-3-Mediated Regulation of the YAP Signaling Pathway in Alveolar Epithelial Cells

The transcriptional activators Yes-associated protein (YAP) and transcriptional coactivator with PDZ-binding motif (TAZ) in the Hippo signaling pathway have been identified as a key regulator in the regeneration and repair of alveolar epithelial cells, as well as in preventing remodeling in fibrosis [101]. In pulmonary fibrosis, abnormal activation of YAP/TAZ is a significant feature of dysregulation of alveolar epithelium. Conversely, YAP/TAZ deficiency also leads to pathological alveolar remodeling and prevents differentiation of AT2 cells to AT1 cells, leading to intra-alveolar collagen deposition and persistent inflammation [101,102].

It has been observed that the P4 ATPase ATP8A1 (an endosomal phosphatidylserine (PS) flippase) localized on recycling endosomes is capable of flipping PS to the cytoplasmic side of the endosomes, which results in the activation of YAP [103,104]. This ATP8A1/PS-regulated YAP signaling pathway has been found to regulate HPS2 pulmonary fibrosis [105]. In AT2 cells, the HPS protein complex AP-3 mediates the transport of ATP8A1 from endosomes to LBs for LB maturation. When the subunit HPS 2 or HPS10 of AP3 is mutated, ATP8A1 is unable to transport to LBs and, thus, accumulates in endosomes, leading to PS being enriched in the cytoplasmic layer of the endosomal membrane, which subsequently causes aberrant activation of YAP and dysfunction of LBs [105]. The aberrant activation of YAP and abnormality of LBs facilitate the fibrotic process of lung alveoli [105]. These findings highlight the importance of AP-3 in mediating LB maturation in AT2 cells and suggest that mutated AP-3 may contribute to the pathogenesis of HPS2/10-associated pulmonary fibrosis by altering alveolar epithelial cell homeostasis upon impairing the transport of key cargoes to LBs.

### 3.2. Immune Dysregulation and Inflammatory Response in HPS-PF

In the lungs of patients with HPS-PF, both honeycombing and granular, glassy turbidity can be observed by high-resolution computed tomography (HRCT) scanning examinations [106]. Furthermore, significant inflammatory features can also be observed in the lungs of mouse models of HPS and in human patients with HPS [23,107]. The immune dysregulation and inflammatory reaction in the lungs were found to be closely related to the development of pulmonary fibrosis [108]. The immune cells, including macrophages, monocytes, neutrophils, mast cells, and lymphocytes, are involved in the progression of HPS-PF. The malfunction of LROs due to HPS gene mutations in these cells may cause fibrotic immunity and inflammation in HPS-PF. The effects of various immune cells on HPS-PF are summarized in Table 2.

#### 3.2.1. Macrophages

Elevation of macrophage numbers has been observed in the bronchoalveolar lavage fluid of the lungs in HPS patients or Hps1 and Hps2 mice [114]. In addition, the macrophages in the Hps model mice are in the form of large foamy cells [115], and macrophages in HPS1 patients show enhanced activity of mTOR and exhibit a distinctive immuno-metabolic profile [27]. Furthermore, over-secretion of nitric oxide synthase and the MCP-1 by the lung epithelium in the Hps mice activates alveolar macrophages, leading to enhanced production of chemokines, inflammatory cytokines (including macrophage inflammatory protein-1Alpha [MIP-1α], MCP-1, and TGF-β) [95,109,114,115]. Overactive macrophages facilitate AT2 cell apoptosis and fibrotic progression [109]. High concentrations of MIP-1α, granulocyte-macrophage colony-stimulating factor (GM-CSF), and macrophage colony-stimulating factor (M-CSF) were found in macrophages in patients with mild HPS1 pulmonary fibrosis [109], and these inflammatory factors may be the pathogenic cause of early HPS1 pulmonary fibrosis. Notably, the concentration of MCP-1 correlated with the severity of pulmonary fibrosis in HPS1 patients [109]. However, the severity of pulmonary disease in HPS2 patients appears to have no significant correlation with levels of MIP-1α, MCP-1, interferon-gamma (IFN-γ), regulated-upon-activation normal T cell expressed and secreted (RANTES), GM-CSF, platelet-derived growth factor BB (PDGF-BB), matrix metalloproteinase-1 (MMP-1), and matrix metalloproteinase-7 (MMP-7) [42]. Instead, it is associated with levels of TGF-β1 and interleukin-17A (IL-17A) [42]. The IL-17RA signaling pathway is necessary for the production of pulmonary TGF-β1, collagen deposition, and fibrosis development [116]. Elevated expression of TGF-β1 may promote the progression of pulmonary fibrosis by activating the epidermal growth factor receptor (EGFR) pathway [117]. In addition, AT2 cells in Hps9 mutant mice also exhibited increased MCP-1, and foamy alveolar macrophages were found in bronchoalveolar lavage fluid from HPS5 patients [118]. These are similar to the pulmonary fibrosis phenotype of HPS1, suggesting that mice or humans with BLOC-1/2 mutations may be at risk of pulmonary fibrosis.

In conclusion, alveolar macrophages are a significant source of cytokines and chemokines. Sustained inflammatory response in HPS may produce large amounts of cytokines and chemokines that promote pulmonary fibrosis. Thus, these cytokines and chemokines could serve as potential molecular markers for monitoring the progression of HPS-PF.

#### 3.2.2. Monocytes

Inflammatory monocytes have been identified in blood samples from patients with HPS1 [27]. HPS1 patients showed high expression levels of CD64 and CD62L with classical CD14 monocyte markers and reduced CD16 in a cluster of cells with CD14 and CD16 intermediate monocyte markers [27]. A comparable phenomenon was observed in patients with AP-3 complex deficiency [111].

Monocytes from HPS1 patients show an inflammatory phenotype linked to dysregulation of IL-1α, tumor necrosis factor (TNF), oncostatin M (OSM) in serum, and monocyte-derived macrophages [27]. In addition, multi-tissue single-cell analysis has shown that the BLOC-3 pathway proteins are highly expressed in myeloid cells [27], suggesting that BLOC-3 and RAB32/38 may participate in the regulation of the inflammatory process. Furthermore, studies have found that the depletion of BLOC-3 or RAB32 limits the ability of macrophages and monocytes to resist bacterial infection [119]. This indicates that monocytes also play a role in the immune dysregulation and inflammatory response in HPS, which may contribute to HPS-PF.

#### 3.2.3. Neutrophils

Neutrophils play a crucial role in HPS-PF in HPS2/4 patients [61]. HPS2 is clinically characterized by severe neutropenia [120,121]. In two patients with HPS2, a notable reduction in neutrophil elastase (NE) content and a considerable elevation in CD63 expression on neutrophils were observed [122]. In addition, mislocalization of granulin–myeloperoxidase (MPO) in neutrophils was observed in HPS2 patients [123]. A recent study identified new cases of HPS2 that show pulmonary infiltrates with high titers of myeloperoxidase-specific anti-neutrophil cytoplasmic antibodies (MPO-ANCA), though typical vasculitis symptoms were not observed [124]. However, in an iPSC model of HPS2 patients, stem cell-derived HPS2 neutrophils have reduced myeloid differentiation and increased uptake by macrophages [122]. These findings indicate that HPS2 may influence the sorting and transport of granule proteins within neutrophils, leading to a decrease in neutrophil production and subsequent inflammation, which may contribute to the progression of HPS-PF.

#### 3.2.4. Mast Cells

One study found that patients with HPS-PF also had mast cell compartment abnormalities [110]. In HPS1 pulmonary fibrosis, lung mast cells are localized in fibrotic areas of the lung parenchyma, and HPS1 dermal mast cells exhibit abnormal granulation, cellular activation, cytokine release, and synthesis of matrix components compared to the healthy controls [110]. Furthermore, in vitro, cultured mast cells from HPS1 patients also have constitutive activation, reduced mediator content, and impaired secretory responses [110]. HuMC cell (human mast cell) granules are known as LROs [5]. Unlike normal HuMC cells, the HPS1 HuMC cells have reduced expression of CD117 and FcεRI, which may cause an increase in CD63 and CD203c (mast cell surface activation marker) levels and a decrease in mediator release, including histamine [110]. The HPS1 HuMC cells release higher concentrations of IL-6, IL-8, galactoglucan lectin-3 (Gal-3), and fibronectin-1 (FN-1) compared to normal HuMC cells [110]. HPS1 transduction rescues these morphological, cytokine, and matrix secretion abnormalities [110]. Furthermore, serum IL-6 has been used to predict early functional decline and death in interstitial lung disease associated with systemic sclerosis [125]. These findings suggest that inflammatory cytokines released by HuMC may promote HPS-PF and serve as targets for therapeutic intervention.

#### 3.2.5. Lymphocytes and Other Immune Cells

The peripheral blood environment of patients with HPS1 pulmonary fibrosis is characterized by activation and population expansion of T and B cells [107]. These patients were found to have high numbers of peripheral blood central memory helper cells, including a significant increase in IgA^+^ memory CD27^+^ B cells, CD38^+^ memory CD27^-^ B cells, IgM^+^ and IgD^+^ B cells, and CD39^+^ helper T cells, whereas CD39^-^ helper T cells and CD4CD25CD127 regulatory T cells exhibited a significant decrease [27,107]. In addition, serum leptin levels were significantly elevated in HPS1 patients [107]. Leptin has been found to stimulate the JAK2/STAT3 and p38MAPK/ERK1/2 signaling, promote the release of B-cell cytokines, and enhance T-cell activation and proliferation [124,126]. Thus, leptin may be a potential target for the treatment of HPS-PF. In conclusion, these immune cell changes may lead to overactivation of the immune system in HPS1 patients, thereby exacerbating the progression of pulmonary fibrosis. Therefore, monitoring the change in immune status is important for HPS-PF treatment.

Unlike HPS 1 patients, significant immunodeficiency was observed in HPS2 patients. Monocyte-derived dendritic cells (moDC) from HPS2 patients had significant defects in maturation and cytokine secretion, resulting in their inability to efficiently secrete interferon alpha (IFN-α) and activate T cells [111]. In addition, natural killer (NK) cells in HSP2 patients are defective in the release of IFN-γ and TNF-α, leading to a diminished ability to attack tumor cells [127]. Recent studies have found that the release of cytokine IFN-γ from CD8^+^ T cells is necessary for alveolar epithelial repair during bacterial pneumonia [128]. Thus, it is speculated that HPS2 patients develop pulmonary fibrosis and tumorigenesis mainly due to immunodeficiency, implying that AP-3 plays a key role in regulating the immune response and maintaining immune homeostasis. Therefore, we should further investigate the role of BLOCs and AP-3 in immune regulation to understand the mechanism by which the immune cells regulate HPS-PF.

### 3.3. Metabolic Reprogramming and HPS-PF

Metabolic reprogramming, as defined by changes in cellular metabolic patterns [129], plays a pivotal role in the pathogenesis of pulmonary fibrosis [130], and represents a promising therapeutic target against pulmonary fibrosis [129,130,131,132]. Metabolic reprogramming or dysregulation induces autophagy [133], apoptosis [134], senescence, and inflammatory responses [135]. All these metabolic reprogramming-induced cellular processes are involved in the pathogenesis of lung fibrosis.

It has been found that macrophages in HPS1 patients display downregulated expression of genes involved in lipid storage and fatty acid metabolism, and cause changes in low-density lipoprotein (LDL) and fatty acid metabolism that lead to the accumulation of LDL and cholesterol [27,136]. In addition, RAB32 in HSP1 macrophages activates the mTOR signaling pathways, which impairs bacterial infection and cytokine production [27]. These findings highlight the significance of changes in cellular metabolic processes in HSP1. Lung biopsies from five patients with HPS identified ceroid-like material in alveolar macrophages [137]. The aggregation of this ceroid-like material is an indication of metabolic storage disorders and a strong association with pulmonary fibrosis [137,138]. Moreover, high levels of lipids were found in dermal fibroblasts of HPS1 and HPS2 patients. However, the accumulated lipids differ between HPS1 and HPS2 genotypes [136,139], suggesting that different genotypes of HPS may have different defective patterns of lipid metabolism. Thus, the accumulation of specific intracellular lipid species in patients with HPS can be utilized as a diagnostic and prognostic marker for HPS-PF.

Airway inflammation and remodeling are associated with aberrant changes in glycolysis and oxidative phosphorylation (OXPHOS) in pulmonary fibrosis [140]. The alveolar epithelium in IPF patients undergoes dramatic metabolic reprogramming with a reduction in mitochondrial oxygen consumption [141]. Similar metabolic reprogramming also occurs in patients with HPS-PF [64]. HPS1-deficient AT2 cells have enhanced mitochondrial fragmentation that correlates with impaired phosphorylation of dynamin-related protein 1 (DRP1) at the serine (Ser) 637 (inactivated phosphorylation) [136,142]. Furthermore, the HPS1 model cell line, the mouse lung epithelial 15 cell line derived from HPS1 lung epithelial cells (MLE15/DHPS1), exhibits robust glycolysis but with significantly reduced mitochondrial respiration, as evidenced by a decrease in oxygen consumption (OCR) and a reduction in total adenosine triphosphate (ATP) levels. It seems that this is linked to a rise in mitochondrial ROS levels and the inhibition of oxidative phosphorylation [142]. The survival of MLE15/DHPS1 cells is dependent on glucose, and death of the cells occurs after activation of the energy sensor 5’ AMP-activated protein kinase (AMPK) and the reduction of glucose in the culture medium [142]. Interestingly, a similar cellular metabolic activity has been found in HPS2 lung epithelial cells [29]. These results suggest that HPS is able to drive metabolic reprogramming, alter the structure and function of mitochondria in alveolar epithelial cells, and promote lung inflammation and fibrosis through multiple pathways.

### 3.4. Aging as a Potential Driver of HPS-PF

Cellular senescence is an irreversible state of growth arrest in which cells exhibit “aging phenotype” characteristics, including instability of the genome, shortening of the telomere, changes in epigenetics, abnormality of proteases, resistance to apoptosis, dysfunctional mitochondria, and defects in autophagy [143]. However, the development of HPS-FP disease is associated with severe surfactant accumulation, lysosomal stress, and AT2 cell apoptosis [144], and it has been found that spontaneous pulmonary fibrosis in double mutant Hps1/2 mice is associated with age, suggesting that pulmonary fibrosis with HPS gene mutation is associated with senescence of AT2 cells [93,94]. Similarly, studies found upregulation of genes related to cellular senescence (for example, β-galactosidase, a marker of senescent cells) detected in AT2 cells in the HPS mouse model [93,94]. Furthermore, in IFP, it was found that it is senescence rather than injury of AT2 cells that leads to progressive pulmonary fibrosis, and that p53 activation is necessary to induce AT2 senescence leading to progressive pulmonary fibrosis [32]. Interestingly, HPS AT2 cells were found to persist in a Krt8 reprogramming transition state mediated by p53 activity [94], a state that promotes alveolar regeneration after lung injury and persists during fibrosis [145].

In addition, mutant lung and isolated lung fibroblasts in HPS1 patients also exhibited an aging profibrotic phenotype, characterized by decreased solute carrier family 7 member 11 (SLC7A11) mRNA expression and increased expression of tissue remodeling genes, such as MMP-2, MMP-9 and TGF-β1 [146]. Reduced expression of SLC7A11 is associated with decreased cellular function and oxidative stress during aging [147]. Following reduced expression of SLC7A11, the plasma cystine (CySS) levels were significantly increased in Hps1 deficient mice, and HPS1 lung fibroblasts showed a decreased ability to recover from oxidative challenge [146]. These data suggest that the onset of HPS1 fibrosis may be due to senescence leading to redox dysregulation which in turn affects the progression of HPS lung fibrosis. However, deletion of BLOC-3 by the gene-editing technique in mouse lung epithelial 12 (MLE12) cells significantly increased the levels of heat-shock proteins (HSPs) associated with the upregulation of various cellular senescence markers, including DNA damage markers, such as cyclin-dependent kinase inhibitor 1 (p21), phosphorylated histone H2AX (γH2AX), and senescence-associated secretory phenotype (SASP) markers, such as CCL2, IL-1α, and IL-6 [148].

In conclusion, HPS1/2/4 deficiency disrupts the normal function of alveolar epithelial progenitor cells and drives cellular senescence. Senescence of alveolar epithelial cells is associated with progression of pulmonary fibrosis. Therefore, restoration of HSPs expression may be effective in improving epithelial cell health, thereby alleviating HPS1 pulmonary fibrosis. In addition, elevating the expression or activity of SLC7A11 may enhance the antioxidant capacity of the cells, thereby reducing HPS1 pulmonary fibrosis.

### 3.5. Lung Fibroblasts and HPS-PF

Fibroblasts are regarded as one of the major effector cells in pulmonary fibrosis diseases. Abnormal activation, proliferation, migration, invasion, and extracellular matrix accumulation of fibroblasts may change the functional structure of alveoli and pulmonary vasculature and cause fibrosis [149]. Thus, lung fibroblasts may play a pivotal role in the progression of HPS-PF.

Lung fibroblasts may communicate with AT2 cells during the pathogenesis of HPS-PF. Using the HPS AT2 and fibroblast organoid culture approach, it was found that the organoid colony formation efficiency (CFE) was significantly reduced in co-cultured AT2 and fibroblast organoids from the lungs of Hps1 and Hps2 mice. Interestingly, normal fibroblasts were able to rescue the organoid CFE of HPS1 AT2 cells, but not HPS2 [150]. Alterations in signaling pathways, including fibroblast growth factor, tensin, fibronectin, angiopoietin-like, and midkine signaling pathways were observed in both co-cultured HPS1 and HPS2 AT2 and fibroblast organoids [150]. HPS1 fibroblasts, but not HPS2 fibroblasts, showed enhanced motility in the scratch assay [150]. According to this, lung fibroblasts isolated from HPS1 patients exhibited enhanced migratory capacity.

Furthermore, silencing HPS1, HPS4, or RAB32 in normal lung fibroblasts enhances p38 protein phosphorylation, and elevates the level of myosin IIB, which is the p38 downstream effector for promoting cell motility, thereby promoting migration of lung fibroblasts [28]. Blockade of the angiotensin receptor AGTR1 by chlorthalidomide reduces myosin IIB level and inhibits in vitro migration of HPS lung fibroblasts [151]. These findings suggest that the BLOC-3 proteins HPS1 or HPS4 play an important role in regulating myosin IIB in lung fibroblasts and may promote fibroblast migration through the AGTR1–p38 MAPK–myosin IIB pathway [28]. Given the deleterious effects of uncontrolled fibroblast migration on lung fibrosis, the interaction of fibroblasts with AT2 cells in HPS may play an important role in the pathogenesis of HPS-PF.

### 3.6. Autophagy and HPS-PF

It has been demonstrated that altered autophagy influences the progression of pulmonary fibrosis by promoting extracellular matrix (ECM) production [152], myofibroblast transformation [153], epithelial-mesenchymal transition [154], epithelial cell dysfunction [155], and inhibiting fibroblast apoptosis [156]. Previous studies have identified that autophagy plays a role in controlling the development of various LROs, such as WPBs [157] and melanosomes [158,159]. Interestingly, autophagy also regulates LB formation and maturation [160]. Atg7 knockout or knockdown mice display abnormal lung morphology and inhibit surfactant protein B production and LB maturation [160]. Autophagy-related protein microtubule-associated protein 1 light chain-3β (LC3B) is localized to the limiting membrane of the LB. However, in HPS1/2-deficient cells, LC3B binds predominantly to the inside of the LB [31], and it appears that HPS1/2-deficiency affects LC3B binding to the limiting membrane of the LB. Furthermore, HPS1 knockdown in Hps1/2 mice and A549 cells results in increased total LC3B, Atg7, Atg5, p62, and TFEB (a regulator of lysosomal biogenesis) [31]. In addition, the severity of pulmonary fibrosis in HPS2 is associated with the level of IL-17A. IL-17A induces mitochondrial dysfunction in AT2 cells by inhibiting PINK1-mediated mitochondrial autophagy, ultimately leading to apoptosis of AT2 cells and promoting fibrosis [99].

Therefore, targeting autophagy may represent a promising therapeutic strategy for HPS-PF.

## 4. Current Therapeutic Strategies for HPS-PF

There are few effective clinical therapeutic drugs or methods for the treatment of HPS-PF. Currently, the therapy for HPS-PF is the same as the treatment of other types of tissue fibrosis. The clinical treatment and potential therapeutic targets and strategies for HPS-PF are summarized in Figure 2.

### 4.1. Current Approved Clinical Drug Therapy for Pulmonary Fibrosis

There are two FDA-approved drugs, nintedanib (Ofev) and pirfenidone (PFD), for the treatment of idiopathic pulmonary fibrosis (IPF) and progressive pulmonary fibrosis currently in the clinic. Nintedanib (Ofev), a tyrosine kinase inhibitor that inhibits multiple receptor tyrosine kinases (RTKs), such as FGFR, VEGFR, PDGFR, CSF1R, and FLT3, was initially developed for restriction of neo-angiogenesis in lung tumor [161,162]. Nintedanib effectively blocks the activation of lung fibroblasts, and, thus, has an inhibitory effect on lung fibrosis and becomes a promising therapeutic drug for pulmonary fibrosis [163]. A recent case report of HPS1 pulmonary fibrosis with lung transplantation after long-term nintedanib administration revealed the potentially beneficial effect of nintedanib on HPS-PF [24]. However, some patients with IPF have a risk of bleeding with the use of nintedanib, and patients with HPS-PF may also have a risk of bleeding due to coagulation disorders. Nevertheless, the drug is deemed a contraindication for HPS-PF.

Pirfenidone (PFD) is a drug that alleviates pulmonary fibrosis by producing an anti-inflammatory effect and modulating the NF-κB, TGF-β, and Wnt signaling pathways [164,165,166]. Administration is prone to mild adverse effects related to gastrointestinal, cutaneous, neurologic, and hepatic function [167]. Pirfenidone reverses and halts the progression of HPS-PF [168] and attenuates the decline in vital capacity (VC) and forced vital capacity (FVC) relative to baseline in patients with HPS-PF [167]. However, the efficacy of the two drugs alone is still suboptimal [24,167].

Recently, it has been demonstrated that the combination of pirfenidone and nintedanib exerts cumulative antifibrotic effects on macrophages and fibroblasts through the SPP1-AKT signaling pathway [169]. This combination drug therapy may offer a promising avenue for the treatment of patients with HPS-PF. It is imperative to identify more targeted drugs and explore combination drug treatment to address the limitations of individual drug treatment to achieve the desired outcomes.

### 4.2. Potential Therapeutic Targets for Treatment of HPS-PF

Currently, very limited therapeutic targets have been identified for treatment of HPS-PF. There are four proteins, namely thromboxane A2 receptor (TBXA2R), cannabinoid receptor type 1 (CB1R), inducible nitric oxide synthase (iNOS), and galactose lectin-3 (Gal-3), involved in inflammation and TGF-b signaling and proliferation of fibroblasts that may be therapeutic targets for treatment of HPS-PF (Figure 2). Targeting these proteins may impede or delay HPS-PF progression.

#### 4.2.1. Thromboxane A2 Receptor

A recent study has identified an increased expression of the thromboxane A2 receptor (TBXA2R) in fibroblasts during lung fibrosis in HPS-PF mice and humans [170]. TBXA2R has been shown to act as a regulator of the TGF-β signaling pathway [170]. F2-isoprostanes (F2-IsoPs) are a marker and mediator of oxidative stress and act as a surrogate TBXA2R ligand [171,172]. It has been demonstrated that F2-IsoPs induce activation of fibroblast proliferation and facilitate myofibroblast differentiation through the TBXA2R-mediated TGF-β signaling pathway [170]. Furthermore, TBXA2R antagonism has been shown to attenuate HPS-PF [170]. These data suggest that TBXA2R may be a promising target for the treatment of HPS-PF. In addition, several TBXA2R antagonists have entered phase II clinical trials. Although they are mainly used to treat allergic asthma, coronary artery disease, and other diseases [173], we expect that the early clinical transformation of TBXA2R signaling pathway therapy will be able to improve the prognosis of HPS-PF patients.

#### 4.2.2. Cannabinoid Receptor Type 1 (CB1R) and Inducible Nitric Oxide Synthase (iNOS)

CB1R and iNOS are overexpressed in the lungs of patients with HPS-PF and bleomycin-induced Hps1 fibrotic mice [174]. The level of endocannabinoids (the CB1R ligands) is elevated and negatively associated with lung function parameters in HPS-PF patients and Hps1 mice [174]. Notably, a recent report discovered that the level of the circulating endocannabinoid anandamide in blood serves as a biomarker for early HPS-PF [161]. It has been shown that CB1R causes abnormal oxidative stress, metabolism, and inflammation by activation of TGF-β signaling [175]. iNOS catalyzes the production of pro-inflammatory reactive nitrogen and plays a role in lung cell damage and apoptosis [114]. The HPS1 deficiency-induced pulmonary fibrosis phenotype can be mediated by the TGF-β -IL-11 axis [98], and iNOS also plays a pivotal role in regulating the expression and secretion of IL-11 in activated lung fibroblasts [174]. In addition, MRI-1867 is a peripheral dual CB1R/iNOS antagonist that eliminates bleomycin-induced increases in amount of profibrotic interleukin-11 in the lung by inhibiting iNOS and reverses mitochondrial dysfunction by inhibiting CB1R [174]. Simultaneous inhibition of both CB1R and iNOS has a greater anti-fibrotic effect compared to targeting either one alone [174]. Thus, the use of MRI-1867 represents an efficacious anti-fibrotic strategy for HPS-PF. A third-generation CB1R antagonist, S-MRI-1867, has recently been developed. S-MRI-1867 has been shown to be an ideal candidate for the treatment of HPS-PF based on its chiral conversion, in vitro and in vivo pharmacokinetic characterization, and physiologically based pharmacokinetic (PBPK) scaling [176].

#### 4.2.3. Galactose Lectin-3 (Gal-3)

Galactose lectin-3 (Gal-3) is a β-galactoside-binding lectin with profibrotic effects. In IPF, it regulates TGF-β1-driven pulmonary fibrosis [177]. Extracellular Gal-3 stimulates apoptosis in epithelial cells, while intracellular Gal-3 promotes the survival and proliferation of fibroblasts, along with the differentiation of myofibroblasts and macrophages [109]. In patients with HPS-PF, there is a significant accumulation of Gal-3 in bronchoalveolar lavage fluid, and the concentration of Gal-3 is correlated with the severity of the condition [178]. Gal-3 is abnormally transported in the extracellular space of Hps1-deficient mice and in the extracellular space of lung fibroblasts and macrophages [179]. These abnormalities contribute to the pathogenesis of HPS-PF.

The effects of Gal-3 are mediated by its interaction with IL-13Rα2 and (chitinase-like protein 1) CHI3L1 and competition with transmembrane protein 219 (TMEM219) for IL-13Rα2 binding. Gal-3 reduces the anti-apoptotic effects induced by CHI3L1 in epithelial cells, while it enhances the macrophage Wnt/β-catenin signaling pathway [179]. The Wnt signaling pathway is unusually active in lung fibrotic tissues, resulting in the generation and secretion of TGF-β1 [180]. The intracellular levels of Gal-3 are regulated by the HPS genes, which are linked to the pathogenesis of HPS-PF. Furthermore, the aberrant accumulation of Gal-3 may be attributed to aberrant intracellular trafficking of Gal-3. In conclusion, BLOC-3 and AP-3 complexes appear to play a pivotal role in the normal intracellular trafficking of Gal-3 [179]. Consequently, further research is needed to explore the functions of Gal-3 and the interaction of Gal-3 with CHI3L1 and its receptors in HPS-PF. Targeting Gal-3 for the treatment of HPS-PF is a promising avenue in future.

### 4.3. Potential Epigenetics-Based Therapeutic Strategy

Epigenetic processes, which are crucial for normal development and maintaining tissue-specific gene expression patterns, involve heritable alterations in gene expression that occur without changes in the DNA sequence [181]. Epigenetics serves as a multifunctional regulator of fibrosis. In recent years, advances in epigenetics, including chromatin remodeling, histone modifications, DNA methylation, ubiquitination, and noncoding RNAs (ncRNAs), have provided further insight into the mechanisms underlying fibrosis [182,183]. In IPF, epigenetic expression changes may regulate the progression of lung fibrosis [184]. For example, MiR-136-5p [185] and miR-21 mediate fibrotic activation of lung fibroblasts [186], LncRNA MIR99AHG inhibits EMT in lung fibrosis through the miR-136-5p/USP4/ACE2 axis [187], in addition, the ubiquitination of YY1 mediated by the E3 ubiquitin ligase NEDD4 plays a significant role in alleviating IPF [188]. Methylation influences fibroblast activation and fibrosis development in the kidney [189]. Curcumin regulates lung extracellular matrix remodeling and mitochondrial function by modulating the miR-29a-3p/DNMT3A axis to attenuate lung fibrosis [190]. Collectively, these findings indicate that targeting epigenetic mechanisms could be a highly effective therapeutic approach for lung fibrosis. However, studies related to the epigenetics of HPS-PF have not been reported, and we should increase research in this area to screen for targets related to HPS-PF.

## 5. Conclusions and Perspective

Hermansky–Pudlak syndrome is a rare genetic disease. HPS-PF is the major cause of mortality in patients with HPS. Despite recent advances in research into the pathogenesis of pulmonary fibrosis caused by the mutation of HPS genes, the specific molecular mechanism and the differential effect caused by mutation of HPS genes on pulmonary fibrosis remain unclear. The pathogenesis of pulmonary fibrosis is primarily attributed to lung AT2 cell dysfunction, immune cell defect, and aberrant lung fibroblast proliferation. Thus, HPS-PF treatment should focus on remedying the dysfunction of AT2 cells and the HPS-associated immune reaction and, most importantly, on correcting the mutation of the HPS genes by gene-editing techniques, such as the CRISPR-Cas9 gene editing technique.

Currently, we face three challenges in the therapy of HPS-PF. First, we lack clinical samples and have limited animal models for clinical research. Due to the rarity of HPS-PF and the tendency of patients to exhibit hemorrhagic qualities, as well as the difficulty of obtaining biopsy samples, current research is constrained by the use of animal models. This is despite the recent development of multifunctional stem-cell-derived alveolar organoid models for clinical studies of the disease. Second, we currently have few diagnostic or prognostic markers of HPS-PF. We need more in vitro and in vivo studies on patient tissues, cells, blood, broncho–alveolar lavage fluid, and other samples to identify diagnostic and prognostic markers. Third, we currently have no curative treatment for HPS-PF. Lung transplantation is the only effective treatment currently, but the scarcity of donor organs and the risk of hand-to-hand transmission restrict the application of lung transplantation for the treatment of HPS-PF.

In the future, we may need to search for specific HPS regulatory proteins in lung cells, particularly in AT2 cells, define the function of HPS proteins in lung fibrosis, and identify new therapeutic targets for treatment of HPS-PF. In addition, exploring immune cell reprogramming and defining the role of inflammation in pulmonary fibrosis will be important to advance our understanding of the pathogenesis of HPS-PF. The development of new targeted therapeutic drugs is the key to improving the therapeutic effect of HPS-PF. Exploring gene editing technology to correct HPS gene mutations may eventually cure the HPS and eliminate the HPS-PF.

## Figures and Tables

**Figure 1 ijms-25-11270-f001:**
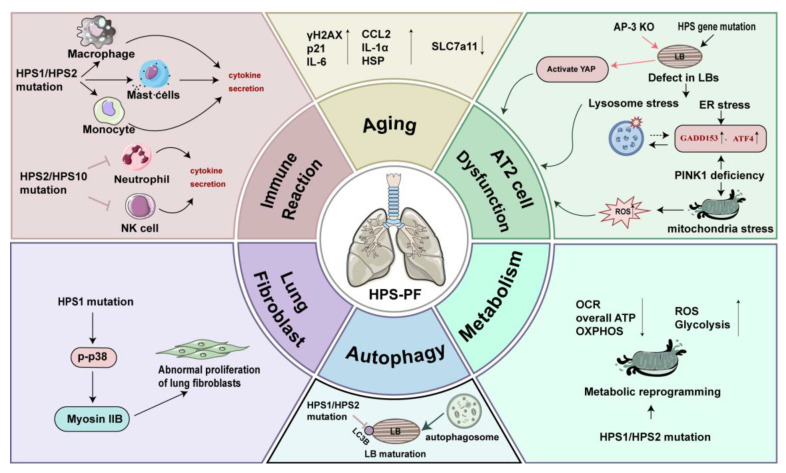
The major pathogenic causes of HPS-PF. Mutations in the HPS genes impact the development of pulmonary fibrosis through multiple pathways. Defective AT2 cells and dysregulated immune reactions caused by mutated HPS proteins along with metabolic dysfunction, abnormal proliferation of lung fibroblasts, autophagy, and cellular senescence promote the progression of HPS-PF. ↑: Upregulation; ↓: Downregulation. AP-3, adaptor protein complex-3; AT2 cell, alveolar type II epithelial cell; ATF4, activating transcription factor-4; ATP, adenosine triphosphate; CCL2, The C-C motif chemokine ligand 2; ER, endoplasmic reticulum; GADD153, growth arrest and DNA damage-inducible gene 153; HPS, Hermansky–Pudlak syndrome; HSP, heat-shock protein; IL-6/1α, interleukin-6/1α; LBs, lamellar bodies; myosin IIB, non-muscle myosin II isoform B; OCR, Oxygen consumption; OXPHOS, oxidative phosphorylation; p21, cyclin-dependent kinase inhibitor 1/CDKN1A; PINK1, PTEN-induced putative kinase 1; ROS, reactive oxygen species; γH2AX, phosphorylated histone H2AX; SLC7A11, solute carrier family 7 member 11; YAP, Yes-associated protein.

**Figure 2 ijms-25-11270-f002:**
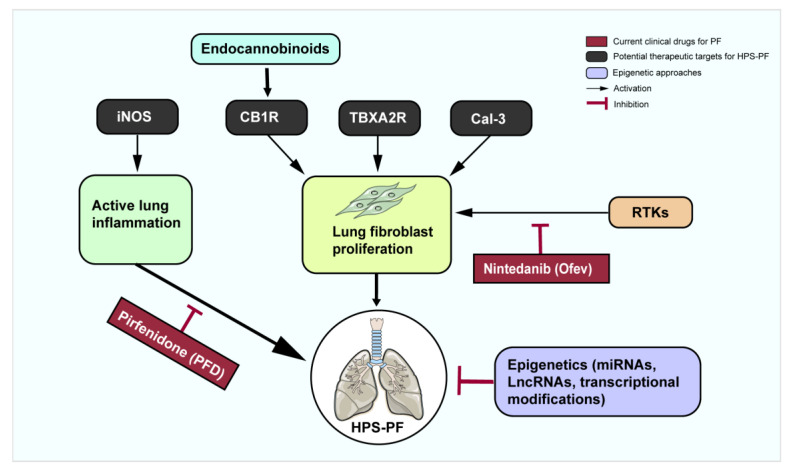
Current therapeutic drugs and potential therapeutic targets for treatment of HPS-PF. Nintedanib (Ofev) and pirfenidone (PFD) are currently two clinically approved drugs for treating pulmonary fibrosis. Nintedanib inhibits receptor tyrosine kinases to suppress fibroblast proliferation, while pirfenidone inhibits lung profibrotic inflammation. Potential therapeutic targets for treatment of HPS-PF, including TBXA2R, CB1R, iNOS, and Gal-3, are shown in the black blocks. In addition, the epigenetic approach is a promising therapeutic strategy for HPS-PF. CB1R, cannabinoid receptor type 1; Gal-3, galactose lectin-3; HPS-PF, Hermansky–Pudlak syndrome-associated pulmonary fibrosis; iNOS, inducible nitric oxide synthase; RTKs, receptor tyrosine kinases; TBXA2R, thromboxane A2 receptor.

**Table 2 ijms-25-11270-t002:** Effects of various immune cells on HPS-PF.

Immune Cell Types	Function	Secreting Cytokines/Chemokines	Association with HPS Pulmonary Fibrosis	
Macrophages	Participate in the inflammatory response, the clearing of pathogens, and tissue repair	HPS1 mutation: MCP-1↑, MIP-1α↑, IL-1Rα↑, IL-8↑, GM-CSF↑, M-CSF↑, PDGF-bb↑, TNF-α↑ HPS2 mutation: TGF-β↑, IL-17A↑	Activation of alveolar macrophages, which oversecrete additional chemokines and inflammatory cytokines to promote fibrosis	[42,95,96,107,109]
Monocytes	Transformed into macrophages and participate in the inflammatory response	HPS1 mutation: TGF-β↑, TNF↑, IL-1α↑, IL-8↑, IL-1Rα↑, OSM↑	HPS1/2 mutation: CD64↑, CD62L↑, CD16↓ Transformation into macrophages to secrete profibrotic factors, in order to enhance local inflammation	[27,107]
Neutrophils	Participate in the acute inflammatory response and clear the pathogens	HPS2/10 mutation: Secretion defects	HPS2/10 mutation: NE↓, MPO↓, MPO-ANCA↑ The reduction in neutrophils may lead to impaired regulation of inflammatory responses, thereby affecting the repair and regeneration process of lung tissue	[61]
Mast cells	Release of histamine and other inflammatory mediators	HPS1 mutation: IFN-γ↑, TNF-α↑, IL-6↑, IL-8↑, FN-1↑, Galectin-3↑, histamine↓	HPS1 mutation: CD63↑, CD203c↑, CD117↓, FcεRI↓ May aggravate the local inflammatory response and the fibrotic process	[107,110]
B cells	Production of antibodies and cytokines	-	HPS1 mutation: IgA^+^ memory CD27^+^ B cells↑, CD38^+^ memory CD27 B cells↑, IgM^+^ and IgD^+^ B cells↑ Play a role in pulmonary fibrosis via immune-memory function	[107]
Treg cells	Maintain immune tolerance and prevent autoimmunity	HPS1 mutation: IL-4↑, IFN-γ↑, TNF-α↑	HPS1 mutation: CD4CD25CD127 Treg cells↓ Immunosuppression imbalance, exacerbating pulmonary fibrosis	[27]
Helper T cells	Activate and direct the responses of other immune cells	-	HPS1 mutation: CD39^+^ helper T cells↑ CD39^-^ helper T cells↓ Dysregulation of the immune response affects the tissue damage and repair process in the pulmonary system	[107]
DC cells	Antigen presentation, and activated T cells	HPS2 mutation: Defects in IL-12, MIP1-β, CXCL9 and INF-α secretion	Dysregulation of the immune response affects the tissue damage and repair process in the pulmonary system	[111]
NK cells	Identify and kill the tumor cells, and then try to kill the infected cells	HPS2 mutation: Defects in TNF- α and IFN- γ secretion	Dysregulation of the immune response affects the tissue damage and repair process in the pulmonary system	[112,113]

↑: Upregulation; ↓: Downregulation.

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
