# Peer review of "Pathogenesis and Therapy of Hermansky–Pudlak Syndrome (HPS)-Associated Pulmonary Fibrosis"

_ijms, 2024, doi:10.3390/ijms252011270_

Round 1

Reviewer 1 Report

Comments and Suggestions for Authors

The topic is complex and difficult to organize in a succinct Introduction, but some aspects are unclear and should be introduced in a clearer, although concise, manner, avoiding repetitions. The addition of a Table summarizing all the genes involved, the complexes affected, and possibly the main pathological outcomes with a focus on PF, should be considered to increase clarity for readers. In addition, in describing previous studies it is often unclear what methodological approach was employed in these studies, and this makes the critical discussion not informative. The section on Therapeutic targets appears not well connected with the rest of manuscript, because these targets appear to emerge from studies on IPF. Authors should better introduce these targets, because they appear poorly related with the molecular mechanisms described in the previous sections.

Major points

The final sentence of the Abstract “This review…facilitates developing new therapeutic interventions” is not correct as a review cannot facilitates new therapeutic interventions and should be reformulated.

In the first paragraph of the Introduction, authors state that HPS “…manifests as ocular or oculocutaneous albinism symptoms” but then write that  “…the 16-bp repeat in exon 15 of HPS1 gene has been linked to widespread hyperpigmentation”. This generates some confusion of the phenotypical features of HPS.

Line 45 “(HPS-PF) is regarded as a major lethal factor in HPS”. Could authors explain what are the other factors and what forms are fatal? Some information is present below only for HPS-1 and -4 and is unclear if all HPS forms may develop, although at different percentages, fatal PF and/or the possible causes of other fatalities.

Line 85 The sentence “In addition, AP-3 and BLOCs play a critical role in the biogenesis and maturation of LROs[39]. BLOCs, including BLOC-1, -2, -3[40].” is unclear, possibly some parts were erroneously deleted.

Some sentences are repetitive, for example at line 92 “HPS progressive pulmonary fibrosis is primarily associated with HPS1, HPS2, and 92 HPS4[39].” and line 50 “Pulmonary fibrosis is closely linked to three genetic variants of Hermansky-Pudlak Syndrome (HPS): HPS1, HPS2, and HPS4.”, as well as lines 97-99 and lines 60-64, or partially lines 125-129.

Line 159 What do authors mean for “early lysosomal stress (induced by cathepsin D)”? Genetic lack of cathepsin? Inactivation of cathepsin D gene by genetic manipulation? Similarly, “that cathepsin D not only facilitates apoptosis of alveolar epithelial cells but also stimulates the proliferation of fibroblasts[32, 169 72, 73]”. What do authors mean, overexpression of cathepsin D?

Also incorrect is the sentence after, i.e. “while ER stress (induced by late ER stress 160 markers C/EBP homologous protein [CHOP/GADD153][69]”, because it confuses markers and inducers

Line 175 The sentence “AT2 cell apoptosis exhibits gene specificity dependent on cysteine asparaginase and is susceptible to fibrosis in single mutant Hps1 and Hps2 mice.” Includes two concepts and their relation is unclear

Line 176 What do authors mean for “spontaneous Hps mice”? And for "healthy" Hps mice? (Line 207)

Line 224 AP-3 is a complex and cannot be “mutated”

Line 269  Cytokines and chemokines are usually generic markers of inflammation, it is unclear how authors imagine they could serve as markers of PF

The section 3.2. Immune dysregulation and inflammatory response in HPS-PF is very long and difficult to follow. Relevant studies involving specific immune cells should be summarized in a Table

Sections 3.2.5 and 3.4, for examples, are useful, but there is poor critical discussion

Minor points

Line 27 “First reported in 1959 To date” is “First reported in 1959, to date…”?

Line 115 alveolar should be Alveolar

Line 154 “and Cardiovascular disease,” is “and cardiovascular diseases.”

Line 214” Abnormal” is “abnormal”

Line 224 Check

Reviewer 2 Report

Comments and Suggestions for Authors

I read with interest the work of Xiao Hu and colleagues.

The paper is accurate, substantial and updated compared to previous reviews on the topic.

Overall, I consider the paper to be of scientific interest and worthy of publication.

Round 2

Reviewer 1 Report

Comments and Suggestions for Authors

The manuscript by Hu et al has satisfactorily addressed issues raised.

A few points to fix in the newly added/thoroughly revised sentences:

In section 3.4 authors state that “Cellular senescence is characterized by an irreversible state of growth arrest in which cells exhibit an “aging phenotype” characteristics, including genomic instability……and autophagy”, but autophagy is not a characteristic, do authors mean autophagy dysfunction or what else?

Line 485/491: Avoid the repetition of “Furthermore"
